# Next-Token Prediction Enables Scalable Learning of Sleep Physiology

## Abstract

Foundation models offer a promising route to compress multi-modal physiological signals into better measures of human health, with broad applications across sleep medicine, cardiology and neurology. Existing models are typically trained with contrastive or masked-reconstruction objectives, both of which have known shortcomings on stochastic, continuous signals. In this work, we show that next-token prediction is a simple and scalable alternative. We develop Hypnos, a multi-modal sleep foundation model trained via next-token prediction over residual-vector-quantization tokens drawn from eight sensing modalities of overnight polysomnography recordings (e.g. EEG, ECG, respiratory signals). A large auto-regressive RQ-Transformer jointly predicts the next token across all modalities in parallel, with a novel modality-masking strategy enabling generalisation to subsets of modalities during inference. Using over 20,000 overnight recordings drawn from nine public datasets, we find that both next-token perplexity and downstream probing performance continue to improve with model scale. Hypnos matches or exceeds prior sleep foundation models and strong supervised baselines on sleep stage classification across in-domain and held-out test sets.

## 1. Introduction

Physiological recordings such as polysomnography (PSG) capture hours of continuous, multi-modal sensor data from the brain and body; a single overnight study with eight channels at 128 Hz yields over 30 million data points. How can we compress hours or even days of such data into better measures of health for tasks such as identifying neurodegenerative or cardiovascular disease? A key motivation for physiological foundation models is to use large quantities of unlabelled sensor data to address this challenge. Prior work has shown that self-supervised learning (SSL) techniques can be used to learn effective representations from a broad range of physiological sensors (Banville et al., 2021; Abbaspourazad et al., 2024; Yuan et al., 2024). These have predominantly been trained using either contrastive learning (Thapa et al., 2026; Abbaspourazad et al., 2024) or masked reconstruction (Lee et al., 2025; Narayanswamy et al., 2024; Fox et al., 2025). However, next-token prediction is a simple and potentially more scalable alternative: it underpins modern Large Language Models (Radford et al., 2019; Brown et al., 2020), scales to context lengths exceeding 1M tokens (Gemini Team, 2024), and has been successfully applied in the analogous audio domain (Borsos et al., 2023; Défossez et al., 2024).

In this paper, we introduce Hypnos, a sleep foundation model trained on eight sensing modalities (EEG, EOG, EMG, ECG, respiratory effort) using residual vector quantization (RVQ) and an auto-regressive RQ-Transformer to jointly predict the next token across modalities. To handle the sensor heterogeneity of clinical recording montages, we also introduce a modality-masking strategy that enables generalisation to arbitrary sensor subsets at inference. Our key contributions are as follows:

- **Large-scale next-token prediction from multi-modal physiological signals:** We demonstrate next-token prediction as an effective SSL objective for multi-modal physiological signals including EEG, ECG and respiratory signals.

- **A state-of-the-art sleep foundation model:** Hypnos outperforms prior sleep foundation models and strong supervised baselines on sleep staging, arousal detection, apnoea detection, and oxygen desaturation, across both in-domain and held-out cohorts.

- **Improved robustness to missing modalities:** A novel modality-masking strategy lets Hypnos generate effective embeddings across common sensor subsets, supporting deployment across device configurations.

---

[1]Anonymous Institution, Anonymous City, Anonymous Region, Anonymous Country. Correspondence to: Anonymous Author <anon.email@domain.com>.

Preliminary work. Under review by the International Conference on Machine Learning (ICML). Do not distribute.

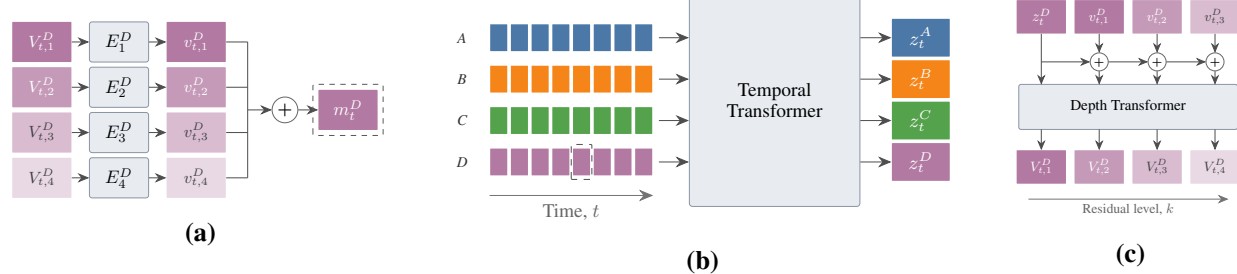

**(a)**      **(b)**      **(c)**

*Figure 1.* **Hypnos model training.** (a) At each time $t$, embeddings from residual tokens of modality $i$ are summed to form an embedding $m_t^i$. (b) The Transformer backbone mixes information across time and modalities to produce embeddings $z_t^i$. (c) The Depth Transformer auto-regressively predicts the next residual token $V_{t,k}^i$ conditioned on $z_t^i$ for all $i, t, k$.

## 2. Method

**Related work** Discrete tokenisation followed by auto-regressive sequence modelling has been used to build foundation models across domains including audio (Borsos et al., 2023; Défossez et al., 2024) and brain signals (Jiang et al., 2025; Xiao et al., 2025). We extend this line of work to multi-modal physiological signals via residual vector quantisation and an RQ-Transformer (Lee et al., 2022a) that jointly predicts multiple high-rate streams in parallel; our technical approach is most similar to Moshi (Défossez et al., 2024), a recent speech-text foundation model. The closest sleep foundation models, SleepFM (Thapa et al., 2026) and OSF (Shuai et al., 2026), train respectively via cross-modal contrastive learning and DINO-style self-distillation (Caron et al., 2021) with random channel masking; BIOT (Yang et al., 2023) uses channel/token dropout augmentations of EEG. We benchmark against SleepFM and OSF as the closest open-source baselines.

**Datasets and modalities** We pre-train and evaluate on overnight polysomnography (PSG) recordings from the National Sleep Research Resource (NSRR, (Zhang et al., 2018)). Following Shuai *et al.* (Shuai et al., 2026), we pre-train on SHHS (Quan et al., 1997), CFS (Redline et al., 1995), CCSHS (Rosen et al., 2003), NCHSDB (Lee et al., 2022b), and WSC (Young et al., 2009), and use MrOS (Song et al., 2015), DOD-H and DOD-O (Guillot et al., 2020) for held-out validation. In total, this corpus contains over 20,000 overnight PSG recordings spanning a broad range of patient demographics, sensors and recording configurations. We use up to $M = 8$ modalities per recording: two central EEG (C3–M2, C4–M1), two EOG (E1–M2, E2–M1), chin EMG, ECG, and abdominal/thoracic respiratory effort. Brain, muscle and cardiac channels are resampled to 128 Hz and respiratory effort to 32 Hz. Full pre-processing details are given in Section B.1, and per-modality channel configurations in Table 3 (Section A.1).

**Tokenization** For each modality we train a tokenizer that maps a 1D channel $X^i \in \mathbb{R}^{f \cdot T}$ into a stream of discrete residual tokens $V^i \in \{1, \ldots, C\}^{K \times T}$, where $f$ is the sampling rate, $T$ the sequence length in seconds, $K$ the number of residual levels, and $C$ the per-codebook size. Our design draws from Mimi (Défossez et al., 2024) and BrainTokenizer (Xiao et al., 2025): an encoder-decoder of stacked SeaNet (Tagliasacchi et al., 2020) and Transformer (Vaswani et al., 2017) layers with a residual vector quantization (Juang & Gray, 1982; Martinez et al., 2014) bottleneck. Strides are chosen so encoders produce embeddings at 1 Hz, and all convolutions are causal. Tokenizers are shared across EEG channels, across EOG channels, and across respiratory channels, giving five tokenizers for eight modalities. We use $K = 8$ residual levels for neural and muscular signals and $K = 4$ for cardiac/respiratory; $C = 2048$. Full architecture, loss formulation, and training hyperparameters are given in Section B.2.

**Hypnos** After tokenization, we use an RQ-Transformer (Lee et al., 2022a) to simultaneously predict the next token of each modality (Figure 1). A learnt embedding layer aggregates the $K$ residual tokens $V_{t,k}^i$ at each $(i, t)$ into a single embedding $m_t^i$. A Transformer backbone then mixes information across time and modalities; for efficiency, each layer applies temporal then modality attention rather than full all-to-all. A depth Transformer auto-regressively predicts residual tokens for the next timestep conditioned on $z_t^i$, with a learnt per-modality embedding added so the shared depth Transformer can disambiguate modalities. Predictions are made in parallel for all residual levels, modalities and timesteps. All Transformer components use causal sliding-window attention (Beltagy et al., 2020). This allows Hypnos to generalise to arbitrary-length recordings during inference. After training we discard the depth Transformer and use the temporal-Transformer outputs $z_t^i$ as embeddings. We evaluate three model scales: **Hypnos-Tiny** (41M), **-Small** (97M), and **-Base** (286M); see Table 4 (Section A.2). Models are trained with AdamW (Loshchilov & Hutter, 2019) and a cosine learning-rate schedule; default batch size $B = 512$, context length $T = 512$ ($\approx 8.5$ minutes

*Table 1.* **Sleep stage classification.** Cohen's $\kappa$, mean over subjects. **Best** in bold; * Hypnos significantly better than both SleepFM and OSF.

| | In-domain | | | Held-out | | |
|---|---|---|---|---|---|---|
| Method | SHHS | CCSHS | CFS | MROS | DOD-H | DOD-O |
| *Supervised* | | | | | | |
| YASA (Vallat & Walker, 2021) | 0.800 | 0.814 | 0.781 | 0.746 | 0.792 | 0.721 |
| U-Sleep (Perslev et al., 2021) | 0.799 | 0.851 | 0.803 | 0.730 | 0.753 | **0.766** |
| SleepTransformer (Phan et al., 2022) | 0.806 | 0.873 | 0.804 | 0.753 | 0.785 | 0.720 |
| *Self-supervised + linear probe* | | | | | | |
| SleepFM | 0.734 | 0.778 | 0.755 | 0.624 | 0.418 | 0.551 |
| OSF | 0.790 | 0.849 | 0.800 | 0.743 | 0.677 | 0.707 |
| Hypnos | 0.807* | 0.875* | 0.837* | 0.797* | 0.796* | 0.717 |
| *Self-supervised + MLP probe* | | | | | | |
| SleepFM | 0.778 | 0.830 | 0.789 | 0.678 | 0.370 | 0.575 |
| OSF | 0.800 | 0.861 | 0.816 | 0.749 | 0.690 | 0.715 |
| Hypnos | **0.816*** | **0.887*** | **0.840*** | **0.805*** | **0.824*** | **0.744*** |

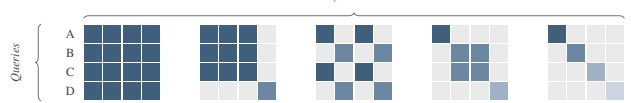

*Figure 2.* **Group masking.** For each training sequence, modalities (e.g. A, B, C, D) are randomly divided into one or more groups. Modalities within a group attend only to each other.

at 1 Hz). All modalities, timesteps and residual levels are weighted equally in the loss. Full hyperparameters are in Section B.3.

**Modality masking** Sleep studies use a wide range of sensor configurations, from full PSG to single-channel EEG or cardio-respiratory wearables. To enable generalisation to arbitrary sensor subsets at inference, we implement a novel group-masking strategy: for each sample we randomly partition modalities into groups by sampling from a Chinese Restaurant process (Aldous, 1985), and modalities within a group attend only to each other (Figure 2). Unlike prior approaches that zero-pad or fully mask modalities (Carter & Tarassenko, 2025; Xu et al., 2025) or use random channel masking as augmentation (Shuai et al., 2026), our partitioning means that all modalities are seen during each forward pass, with multiple distinct groups per training example.

**Embedding aggregation and evaluation** To produce a summary vector $z_t$, we average $z_t^i$ over available modalities; for time-range embeddings we further average over the range. To evaluate the resulting embeddings, following (Shuai et al., 2026) we formulate downstream tasks (sleep staging, arousal/apnoea/desaturation detection) over 30-second windows of data and benchmark Hypnos against SleepFM (Thapa et al., 2026) and OSF (Shuai et al., 2026) as well as supervised baselines (YASA (Vallat & Walker, 2021), U-Sleep (Perslev et al., 2021), SleepTransformer (Phan et al., 2022) via SLEEPYLAND (Rossi et al., 2025)). All probes are trained on the pre-training datasets only, leaving MrOS, DOD-H and DOD-O as fully external test sets.

## 3. Experiments

**Sleep staging** Table 1 compares Hypnos against SleepFM (Thapa et al., 2026), OSF (Shuai et al., 2026), and strong supervised baselines on five-class AASM sleep stage classification. We report Cohen's $\kappa$ averaged over recordings and use a Wilcoxon signed-rank test with Benjamini–Hochberg FDR correction ($q < 0.05$) to assess significance; the same procedure is used for all foundation-model comparisons. With either a linear or MLP probe, Hypnos outperforms SleepFM and OSF with statistical significance on all six datasets, and exceeds the supervised baselines on five of six datasets evaluated. We note this performance was achieved with simple averaging-based aggregation; full fine-tuning or more sophisticated probe heads (*e.g.,* an LSTM as in (Thapa et al., 2026)) are likely to yield further gains.

**Additional sleep tasks and robustness** Hypnos also outperforms SleepFM and OSF on cortical arousal detection, apnoea detection, and oxygen desaturation across SHHS (in-domain) and MrOS (held-out), with statistical significance on most evaluations and matching performance on the rest (Table 2). It is similarly more robust to missing modalities under two clinically motivated subsets (single-channel EEG; cardio-respiratory only), outperforming SleepFM and OSF on every reduced-modality task on held-out MrOS (Table 5 in Section A.3).

**Few-shot learning** Figure 3 reports sleep staging performance as we scale the proportion of recordings used to train the MLP probe from 1% to 100%. Hypnos outperforms SleepFM and OSF across all data fractions; on held-

*Table 2.* **Additional sleep tasks.** AUROC / AUPRC, mean over subjects. **Best** in bold; * Hypnos significantly better than both SleepFM and OSF.

| | SHHS | | MrOS | |
|---|---|---|---|---|
| | AUROC | AUPRC | AUROC | AUPRC |
| *Arousal detection* | | | | |
| SleepFM | 0.927 | 0.766 | 0.882 | 0.695 |
| OSF | **0.951** | 0.830 | 0.922 | 0.777 |
| Hypnos | 0.950 | **0.831** | **0.934*** | **0.808*** |
| *Apnoea detection* | | | | |
| SleepFM | 0.836 | 0.695 | 0.742 | 0.381 |
| OSF | 0.876 | 0.762 | 0.775 | 0.392 |
| Hypnos | **0.887*** | **0.782*** | **0.795*** | **0.418*** |
| *Oxygen desaturation* | | | | |
| SleepFM | 0.779 | 0.479 | 0.738 | 0.719 |
| OSF | 0.790 | 0.502 | 0.737 | 0.715 |
| Hypnos | **0.872*** | **0.659*** | **0.767*** | **0.755*** |

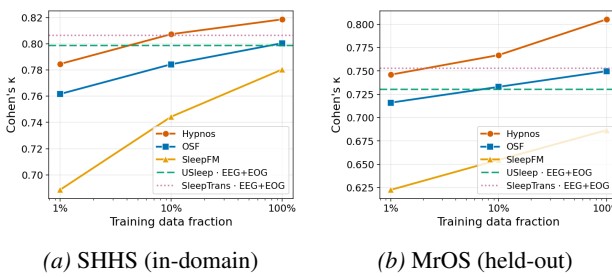

*(a)* SHHS (in-domain)    *(b)* MrOS (held-out)

*Figure 3.* **Few-shot sleep stage classification.** MLP probes trained on varying fractions of in-domain data. With 1% of recordings, Hypnos already matches U-Sleep trained on the full dataset on held-out MrOS.

out MrOS, Hypnos with a probe trained on as little as 1% of recordings already matches U-Sleep trained on the full dataset.

**Model scaling**    Figure 4 shows the effect of scaling Hypnos from Tiny to Base. Both next-token perplexity and downstream probing performance improve monotonically with model size on all three tasks evaluated, supporting next-token prediction as a scalable objective for physiological signals. Further scaling experiments, including to larger model sizes and to context lengths exceeding an hour ($T = 4096$) using unimodal model variants, are reported in Sections C.2 and C.3.

## 4. Conclusions and Future Work

We have presented Hypnos, a multi-modal sleep foundation model trained with next-token prediction over residual vector quantised tokens drawn from eight physiological modalities. Hypnos outperforms prior sleep foundation models on sleep staging, arousal detection, apnoea detection and oxygen desaturation, and matches or exceeds strong super-

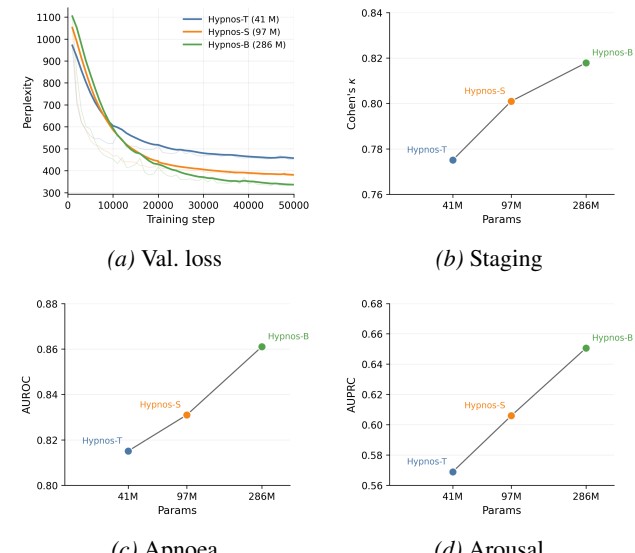

*(a)* Val. loss    *(b)* Staging

*(c)* Apnoea    *(d)* Arousal

*Figure 4.* **Scaling Hypnos.** Next-token perplexity and downstream metrics (linear probe, SHHS) improve monotonically with model scale.

vised baselines on sleep staging across in-domain and held-out cohorts. Next-token perplexity and downstream probe performance improve smoothly with model scale, and a novel modality-masking strategy lets a single trained model generate effective embeddings for different sensor subsets, supporting deployment across diverse device configurations.

Important directions for future work include scaling next-token prediction to longer contexts (hours to days) to capture circadian and multi-night structure, extending evaluations to ambulatory and wearable recordings, and using Hypnos for data-driven biomarker discovery (e.g. identifying latent directions predictive of incident neurodegenerative or cardiovascular disease). Hypnos is one step toward foundation models that compress hours to days of multi-modal physiological signals into better measures of human health; that this simple recipe scales, transfers, and is robust to diverse sensors suggests it can extend beyond sleep.

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

## A. Supplementary figures and tables

### A.1. Modalities used during pre-training

*Table 3.* **Modalities used during pre-training**. Voltage signals are referenced to the contralateral mastoid or derived bipolarly (EMG). Respiratory effort signals are used directly.

| Modality | Rate (Hz) | Tokenizer group | Residual levels, $K$ |
|---|---|---|---|
| EEG (C3–M2) | 128 | EEG | 8 |
| EEG (C4–M1) | 128 | EEG | 8 |
| EOG (E1–M2) | 128 | EOG | 8 |
| EOG (E2–M1) | 128 | EOG | 8 |
| Chin EMG | 128 | EMG | 8 |
| ECG | 128 | ECG | 4 |
| Abdominal effort | 32 | Resp | 4 |
| Thoracic effort | 32 | Resp | 4 |

### A.2. Hypnos model configurations

*Table 4.* **Hypnos model configurations**. Parameter counts include token embedding tables, both Transformer stacks, and per-codebook output heads.

| | Temporal Transformer | | | Depth Transformer | | | |
|---|---|---|---|---|---|---|---|
| Model | Layers | Hidden $D$ | Heads | Layers | Hidden $D$ | Heads | Params |
| Hypnos-Tiny | 12 | 192 | 3 | 4 | 128 | 2 | 41M |
| Hypnos-Small | 12 | 384 | 6 | 4 | 192 | 3 | 97M |
| Hypnos-Base | 12 | 768 | 12 | 4 | 384 | 6 | 286M |

### A.3. Robustness to missing modalities

We further evaluate Hypnos under two clinically motivated sensor configurations: single-channel EEG ($M = 1$), and cardio-respiratory signals (ECG, ABD, THX; $M = 3$). Each configuration is evaluated across all four downstream tasks on held-out MrOS.

*Table 5.* **Robustness to missing modalities.** MLP probe results on held-out MrOS dataset under restricted-modality configurations. **Best** per metric in bold; * indicates Hypnos is significantly better than both SleepFM and OSF.

| | | Staging | Arousal | | Apnoea | | Desat. | |
|---|---|---|---|---|---|---|---|---|
| Setting | Method | $\kappa$ | AUROC | AUPRC | AUROC | AUPRC | AUROC | AUPRC |
| EEG only | SleepFM | 0.588 | 0.835 | 0.620 | 0.659 | 0.292 | 0.610 | 0.608 |
| ($M$=1) | OSF | 0.689 | 0.906 | 0.749 | 0.659 | 0.291 | **0.643** | 0.620 |
| | Hypnos | **0.780**$^*$ | **0.922**$^*$ | **0.790**$^*$ | **0.723**$^*$ | **0.351**$^*$ | 0.635 | **0.628**$^*$ |
| Cardio-resp. | SleepFM | 0.458 | 0.832 | 0.609 | 0.722 | 0.352 | 0.735 | 0.715 |
| ($M$=3) | OSF | 0.541 | 0.849 | 0.632 | 0.762 | 0.379 | 0.742 | 0.722 |
| | Hypnos | **0.644**$^*$ | **0.878**$^*$ | **0.684**$^*$ | **0.797**$^*$ | **0.417**$^*$ | **0.795**$^*$ | **0.777**$^*$ |

## B. Additional Implementation Details

### B.1. Preprocessing

**Referencing and filtering**   EEG and EOG channels were re-referenced against the contralateral mastoid (C3–M2, C4–M1 for EEG; E1–M2, E2–M1 for EOG). Chin EMG was derived bipolarly from the chin electrode pair. ECG and respiratory effort (ABD, THX) were used directly. All signals were resampled using a polyphase filter with an anti-aliasing low-pass.

Brain, muscle, and cardiac channels were resampled to 128 Hz. Respiratory effort signals (ABD, THX) were resampled to 32 Hz, reflecting their lower frequency content. All channels were notch-filtered to suppress mains interference. Per-modality bandpass filters were then applied:

- EEG, EOG, EMG: 0.5–45 Hz bandpass.

- ECG: 0.05–45 Hz bandpass.

- ABD, THX: 0.05 Hz high-pass.

**Normalisation and amplitude compression**  For each channel of each recording we performed winsorized z-score normalisation, clipping at the 1st and 99th percentiles of the signal to ignore extreme outliers when calculating statistics. After normalisation, values lying beyond $\pm 8\sigma$ were log-compressed (rather than hard-clipped) past a knee of $8\sigma$, so that large artefacts remain order-preserving without dominating the reconstruction loss. Originally, the log-compression choice was made to improve adversarial training, with the aim of preventing the discriminator using clipping to identify real samples. We expect hard-clipping would give very similar downstream performance for non-adversarial tokenizers, including our final design.

### B.2. Tokenizer training details

**Architecture**  Each tokenizer consists of a SeaNet and Transformer components plus an RVQ bottleneck, as illustrated in Figure 5. SeaNet stride ratios are chosen so that the product of strides corresponds to one second of input samples, producing tokens at 1 Hz at every modality's sampling rate. The encoder right-pads by one hop length and drops the first warm-up token, so each output token is right-aligned to its segment boundary.

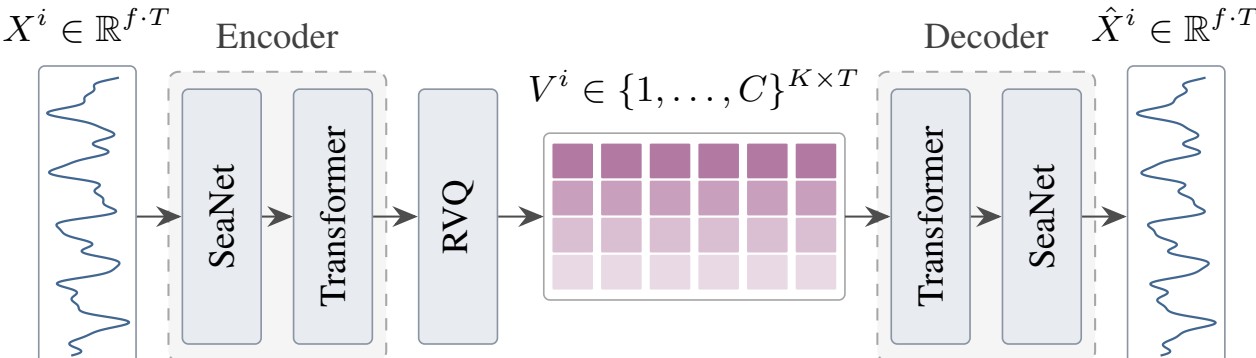

*Figure 5.* **Signal tokenization.** Each single-channel signal $X^i$ of length-$f \cdot T$ is encoded using residual-vector-quantization (RVQ) into a $K \times T$ grid of discrete residual tokens $V^i$. Each residual layer $k$ uses a discrete codebook of size $C$. During training, the encoder and decoder are trained to produce reconstructed signals $\hat{X}^i$, minimising a combination of reconstruction losses and auxiliary losses that encourage high codebook utilisation and stability.

All other architecture and optimisation hyperparameters are listed in Table 6; weight decay is applied to Transformer parameters only, following the Mimi convention (Défossez et al., 2024).

**Loss formulation**  Following BrainTokenizer (Xiao et al., 2025), the tokenizer is trained with a multi-term reconstruction loss combined with an RVQ commitment penalty. The reconstruction loss has (i) a time-domain term between the original waveform $X$ and the reconstruction $\hat{X}$:

$$\mathcal{L}_{\text{time}} \;=\; \|X - \hat{X}\|_1, \tag{1}$$

where $\|\cdot\|_1$ denotes the $\ell_1$ distance; and (ii) a frequency-domain term between the amplitude spectra $A$, $\hat{A}$ and phase spectra $\Phi$, $\hat{\Phi}$ of the Hamming-windowed signals:

$$\mathcal{L}_{\text{freq}} \;=\; \|A - \hat{A}\|_1 + \lambda_\phi \cdot \|\Phi - \hat{\Phi}\|_1. \tag{2}$$

*Table 6.* **Tokenizer hyperparameters.**

| Component | Setting |
|---|---|
| SeaNet `n_filters` | 64 |
| SeaNet stride ratios | $[2, 4, 4, 4]$ at 128 Hz; $[4, 4, 2]$ at 32 Hz |
| SeaNet residual blocks | 1 per stage, dilation base 2 |
| Encoder embedding dim | 512 |
| Codebook entry dim | 256 |
| Codebook size $C$ | 2048 |
| Residual levels $K$ | 8 (EEG, EOG, EMG); 4 (ECG, respiratory) |
| EMA decay | 0.99 |
| Quantization dropout | 0.5 |
| Encoder/decoder Transformer | 4 layers, 8 heads, FFN dim 2048, sliding window 32 |
| LayerScale init | $10^{-2}$ |
| Loss weights | $\lambda_\phi = 0.5$, $\lambda_{\text{RVQ}} = 0.25$ |
| Optimiser | AdamW, lr $2 \times 10^{-4}$, wd $10^{-2}$ (Tx params) |
| Gradient clipping | 1.0 (norm) |
| Schedule | Cosine, 500-step linear warm-up |
| Training steps | 50,000 |
| Batch size $\times$ window | $1024 \times 64\,\text{s}$ |
| Precision | bf16 mixed |

Letting $z_k$ and $z_{q_k}$ denote the residual entering the $k$-th codebook and its nearest entry, the RVQ commitment loss is:

$$\mathcal{L}_{\text{rvq}} \;=\; \sum_{k=1}^{K} \left\| z_k - \text{sg}[z_{q_k}] \right\|_2^2, \tag{3}$$

where $\text{sg}[\cdot]$ denotes a stop-gradient. This term pulls the encoder output $z_k$ toward its assigned codebook entry. Each tokenizer is trained to minimise:

$$\mathcal{L}_{\text{tok}} \;=\; \mathcal{L}_{\text{time}} + \mathcal{L}_{\text{freq}} + \lambda_{RVQ} \cdot \mathcal{L}_{\text{rvq}}, \tag{4}$$

with scalar weights $\lambda_\phi = 0.5$ and $\lambda_{RVQ} = 0.25$, identical to BrainTokenizer (Xiao et al., 2025).

## B.3. Hypnos hyperparameters and training

The temporal Transformer applies rotary position embeddings (RoPE, (Su et al., 2024)) to attention queries and keys, while the depth Transformer uses a learnt position embedding indexed by codebook level. Linear and embedding parameters are initialised from $\mathcal{N}(0, 0.02^2)$ following GPT-2 (Brown et al., 2020). We use a residual dropout of 0.1. Per-scale learning rates are set inversely to hidden width: $1.2 \times 10^{-3}$ for Hypnos-Tiny, $6 \times 10^{-4}$ for Hypnos-Small, and $3 \times 10^{-4}$ for Hypnos-Base. All other training hyperparameters are listed in Table 7.

## B.4. Compute usage

Each tokenizer described in Section 2 was trained using a single NVIDIA H100 GPU using bf16 mixed-precision, requiring 60 GB of GPU RAM and around 5 hours of training time. After training, tokenization was performed using a single NVIDIA L40S GPU, with each entire overnight recording of each channel (10+ hours) tokenized in a single forward pass in around 250 ms. All Hypnos models were trained using H100 GPUs with bf16 mixed-precision training and activation checkpointing in each Transformer layer. Training Hypnos-Base to 100k steps required 3 days distributed across 8 GPUs, using around 45 GB of RAM on each GPU. For downstream probing evaluations, embeddings were generated using a single NVIDIA L40S GPU, with the embedding of each overnight recording taking around 3 seconds.

To reduce compute usage, many of our experiments including initial model design and ablation studies were performed using a unimodal variant of Hypnos-Small using EEG data, which took around 1 day to train on a single H100 GPU. Across all experiments our total compute usage was approximately 4000 H100 GPU-hours.

*Table 7.* **Hypnos training hyperparameters.**

| Component | Setting |
| --- | --- |
| Norm / activation | LayerNorm + SwiGLU; RMSNorm on QK |
| Position encoding | RoPE (temporal); learnt level-index embedding (depth) |
| Sliding-window pattern | Local window 64; every 4th layer global ($T/2$) |
| CRP concentration | $\alpha = 1.0$ |
| Dropout | 0.1 |
| LayerScale init | $10^{-2}$ |
| Weight init | $\mathcal{N}(0, 0.02^2)$ for Linear / Embedding |
| Optimiser | AdamW, $(\beta_1, \beta_2) = (0.9, 0.95)$, wd 0.1 |
| Gradient clipping | 1.0 (norm) |
| Per-scale lr | Tiny $1.2 \times 10^{-3}$; Small $6 \times 10^{-4}$; Base $3 \times 10^{-4}$ |
| Schedule | Cosine, 500-step linear warm-up |
| Training steps | 100,000 |
| Batch size | 512 |
| Context length | 512 |
| Precision | bf16 mixed, activation checkpointing |

## C. Additional Experiments

### C.1. Tokenizer design

**Residual depth** BrainOmni uses a codebook with $K = 4$ quantization layers, which leads to visibly smoothed reconstructions of EEG data (see Fig. 4 of (Xiao et al., 2025)). Instead, we used $K = 8$ quantizers for neural signals to increase reconstruction accuracy of higher frequency details, e.g. gamma activity. Here we investigate the effect of residual depth on downstream performance. We vary the quantization depth at both the input and output to Hypnos, $K_{in}$ and $K_{out}$, which determine the residual tokens available for sequence modelling (Figure 1a) and the residual tokens to be predicted (Figure 1c) respectively. Figure 6 shows the performance of unimodal Hypnos variants across downstream tasks using EEG data as we vary $K_{in}$ and $K_{out}$. We generally observe a slight improvement in performance as we increase $K_{in}$, but a decrease in performance as we increase $K_{out}$. The performance benefit from increasing $K_{in}$ suggests that high frequency input information is beneficial during sequence modelling. The decrease in performance from increasing $K_{out}$ suggests that $K_{out}$ could be decreased during training. Since the Depth Transformer processes $B \cdot M \cdot T$ sequences of length $K$, reducing $K_{out}$ from 8 to 2 would reduce Depth Transformer FLOPs by 75%, and overall FLOPs by around 25% for Hypnos-Base.

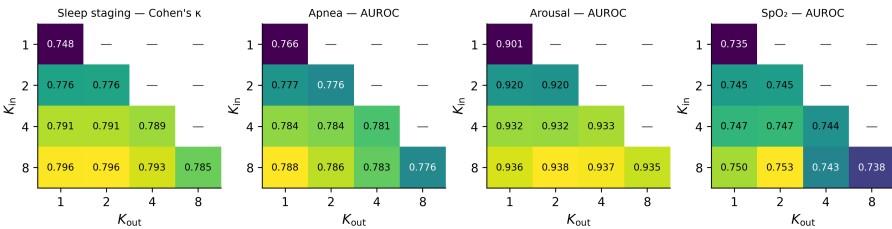

*Figure 6.* **Effect of input and output residual depth on downstream performance.** Linear probe metrics for unimodal EEG models on the SHHS validation set varying $K_{in}$ and $K_{out}$. Performance slightly improves when increasing the number of input residual tokens but worsens when increasing the number of output residual tokens.

**Tokenization length** In our main experiments, we designed our tokenizers to produce tokens at a rate of 1 Hz. This also determines the rate at which unique output embeddings are produced by the model. 1 Hz is a natural choice for real-world applications, aligning with standard units. Additionally, relevant physiological events such as heartbeats and sleep spindles (Andrillon et al., 2011) commonly occur on this timescale. Here we investigate the sensitivity of our approach to the tokenization length of different signals. For EEG and ECG signals, we trained 5 tokenizers with tokenization lengths, $\tau \in (0.25, 0.5, 1.0, 3.0, 5.0)$ seconds. This was achieved by modifying the convolutional stride lengths in the convolutional components of the tokenizers. For 3-second and 5-second tokenization lengths, we also increased the input sequence length

by 3x and 5x respectively, so that the number of tokens observed by the Transformer components in the tokenizers remained constant. We then re-tokenized all datasets and investigated the effect of tokenization length on downstream performance. In Figure 7, we see that increasing tokenization length leads to a decrease in signal-to-noise ratio as the compression rate increases. However, $\tau = 0.25$s (which gives the highest reconstruction SNR) consistently leads to lowest downstream performance. We observed that a token duration of 1-second works well across tasks and inputs.

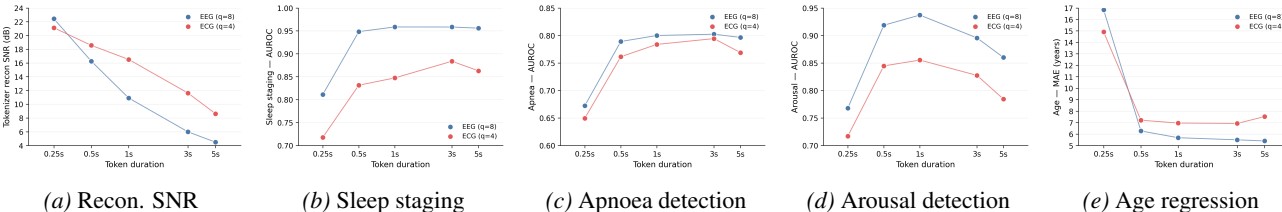

    *(a)* Recon. SNR      *(b)* Sleep staging      *(c)* Apnoea detection      *(d)* Arousal detection      *(e)* Age regression

*Figure 7.* **Effect of token duration on downstream performance.** (left) Reconstruction quality decreases as token duration is varied from 0.25 s to 5 s, i.e. the compression rate increases. However, performance is worst at high token rates (0.25 s) and saturates or regresses beyond 1 s. We adopt a 1 s token duration in all subsequent experiments.

**Adversarial losses** Défossez *et al.* (Défossez et al., 2024) recently observed that removing reconstruction losses and solely relying on adversarial losses led to better performance in downstream audio modelling tasks. In early experiments, we tried incorporating adversarial losses but found that this did not have a significant effect on downstream performance. Additionally, it significantly increased computational requirements: each training run required substantially higher activation memory for the discriminator; and, more training runs were required to find stable hyper-parameters given the well-known instability issues in adversarial training (Salimans et al., 2016).

### C.2. Scaling unimodal EEG models

Compute requirements precluded scaling our multimodal model beyond Base size in our main experiments. To extend the scaling analysis to larger models, we trained unimodal EEG variants of Hypnos from Tiny up to Large. Figure 8 shows next-token perplexity and downstream probing performance on the SHHS validation set as we scale model size. Trends mirror those observed in the multimodal setting: validation loss and downstream metrics continue to improve with scale through to Large.

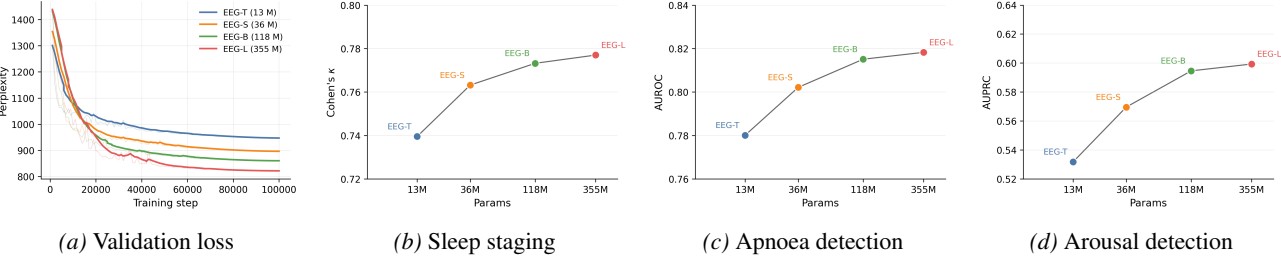

    *(a)* Validation loss       *(b)* Sleep staging       *(c)* Apnoea detection       *(d)* Arousal detection

*Figure 8.* **Scaling unimodal EEG models from Tiny to Large.** Next-token perplexity and downstream metrics continue to improve with model scale. Sleep stage classification, apnoea detection and arousal detection performance are reported using a linear probe on the SHHS validation set.

### C.3. Scaling context length

A key motivation for next-token prediction is that it naturally scales to longer context lengths. To quantify the effect of context length on Hypnos, we trained unimodal EEG and ECG variants of Hypnos-Small with context lengths $T \in \{128, 256, 512, 1024, 2048, 4096\}$ tokens, corresponding to roughly 2 minutes up to over an hour of data at 1 Hz. Models were trained for 50k steps, whilst all other hyper-parameters were held fixed across runs. Figures 9 and 10 show validation perplexity and downstream probing performance on the SHHS validation set as we vary $T$.

For both modalities, validation perplexity improves monotonically with context length, and most downstream metrics continue to improve out to 4096 tokens. Summary tasks benefit most: age regression, CVD risk and obstructive sleep apnoea

(OSA) classification all improve steadily across the full range. Meanwhile, sleep staging and arousal detection saturate earlier, at around 1024–2048 tokens. These trends are consistent across EEG and ECG, suggesting that the benefit of longer context is not specific to a single modality.

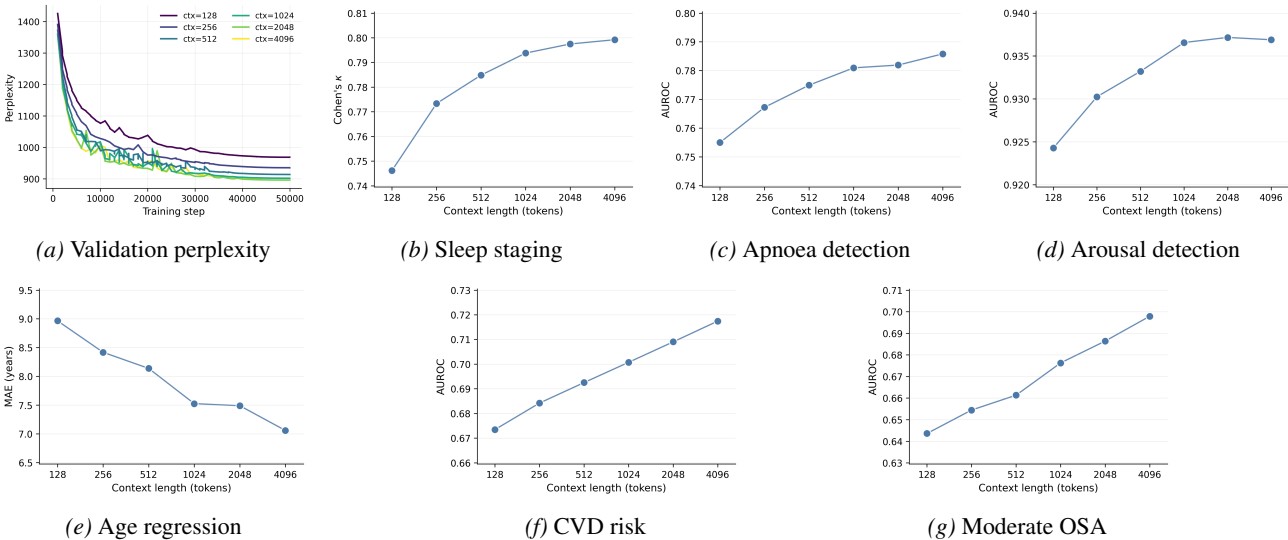

*(a)* Validation perplexity     *(b)* Sleep staging     *(c)* Apnoea detection     *(d)* Arousal detection

*(e)* Age regression     *(f)* CVD risk     *(g)* Moderate OSA

*Figure 9.* **Effect of context length on single-channel EEG models.** Validation perplexity decreases and downstream probing performance improves as the training context length is increased from 128 to 4096 tokens. Sleep staging and arousal detection saturate at around 1024–2048 tokens, while age regression, CVD risk and moderate OSA detection continue to improve at the longest context lengths.

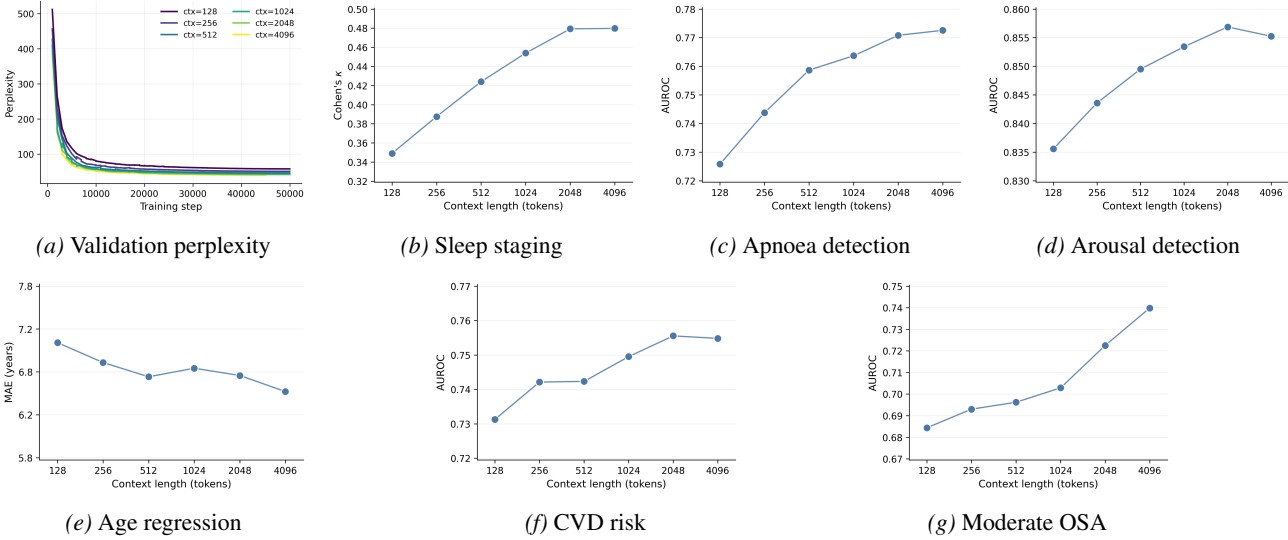

*(a)* Validation perplexity     *(b)* Sleep staging     *(c)* Apnoea detection     *(d)* Arousal detection

*(e)* Age regression     *(f)* CVD risk     *(g)* Moderate OSA

*Figure 10.* **Effect of context length on ECG-only models.** As with EEG, perplexity and downstream metrics improve with longer context. The largest relative gains are again on summary tasks such as moderate OSA detection and CVD risk.

## C.4. Training stability

To reduce compute usage, our main pre-training experiments were performed with a single random seed (42). We observed stable and predictable training dynamics across our training runs including our model scaling experiments. To further assess variance due to neural network training dynamics, we re-trained Hypnos-Small on single-channel EEG using three different random seeds. In Table 8 we see that the variance in validation loss and probing metrics is low.

*Table 8.* **Training stability of Hypnos trained on single-channel EEG across three random seeds.** We report mean ± std and coefficient of variation (CV) across seeds. Downstream task performance is computed on validation data using an online linear probe.

| Metric | Mean ± std | CV (%) |
|---|---|---|
| Validation perplexity (in-domain) | $757.27 \pm 0.50$ | 0.066 |
| Validation perplexity (MrOS held-out) | $876.73 \pm 0.68$ | 0.078 |
| Sleep staging $\kappa$ (SHHS) | $0.7930 \pm 0.0004$ | 0.050 |
| Sleep staging $\kappa$ (MrOS held-out) | $0.7646 \pm 0.0013$ | 0.170 |
| Apnoea AUROC (SHHS) | $0.7825 \pm 0.0014$ | 0.179 |
| Apnoea AUROC (MrOS held-out) | $0.7812 \pm 0.0095$ | 1.216 |

