# OpenReview forum: "Next-Token Prediction Enables Scalable Learning of Sleep Physiology"
_ICML.cc/2026/Workshop/FMSD — FMSD @ ICML 2026 Poster_

### Official Review · Reviewer_ZW8t · 2026-05-18
**Review of Next-Token Prediction Enables Scalable Learning of Sleep Physiology**

**Rating:** 8
**Confidence:** 4

**Review:**

## Summary

This paper introduces Hypnos, a multimodal sleep physiology foundation model trained with next-token prediction over RVQ-tokenized polysomnography signals. The paper reports strong performance on sleep staging and related tasks, including in-domain and held-out datasets.

---
## Strengths

1.	The paper is well motivated: next-token prediction is a natural alternative to contrastive or masked-reconstruction objectives for long physiological recordings.
2.	The architecture is technically interesting. The use of SEANet/Transformer tokenizers with RVQ provides a bridge between continuous physiological data and language-model-style token prediction.
3.	The empirical results are strong, especially on sleep staging, where Hypnos outperforms SleepFM and OSF in most reported settings.

---
## Areas for Improvement

1.	Montage/generalization clarity.
The model uses predefined sleep-PSG derivations such as C3–M2 and C4–M1, with shared EEG tokenizers. The paper should clarify whether it can handle different EEG montages/references, or whether those channels must be treated as missing.
2.	Missing-modality vs. montage robustness.
Modality masking supports missing predefined sensor streams, but it does not necessarily solve electrode-location or rereferencing mismatch. The claim of generalization to arbitrary sensor subsets should be softened or clarified.
3.	Pretraining/evaluation overlap.
Some evaluation datasets are also used during pretraining. The paper labels them as in-domain, but should clearly state whether downstream test subjects or recordings were included in self-supervised pretraining.
4.	Baseline fairness and preprocessing.
The paper should clarify whether SleepFM, OSF, and supervised baselines were re-run with the same channel selection, rereferencing, filtering, resampling, and splits, since EEG preprocessing can substantially affect performance.
5.	Reproducibility.
It is unclear whether Hypnos will be released. Without this, reproducing the pipeline would be difficult.

---
## Detailed Comments

1.	See above
2.	The results show that next-token prediction works well, but not that it is better than contrastive or masked reconstruction, because the comparison is not controlled for architecture, data, compute, and preprocessing.

---
## Justification of Score
Top 50% of accepted papers, clear accept. The paper is technically strong, relevant to the workshop, and empirically impressive. My main concerns are about preprocessing fairness, montage generalization, possible pretraining/evaluation overlap, and reproducibility. These issues do not invalidate the contribution, but they should be clarified to make the claims more precise and trustworthy.

---

### Official Review · Reviewer_sy9X · 2026-05-18
**Submission 74 Review**

**Rating:** 6
**Confidence:** 5

**Review:**

## Summary

The paper proposes a foundational sleep model using an autoregressive reconstruction objective in a discretized token space. The authors conduct extensive evaluations across a wide variety of downstream tasks and empirically compare their approach against several state-of-the-art sleep foundation models. Additionally, the paper includes a thorough scaling analysis of the proposed architecture, providing valuable insights into model capacity, data scale, and performance trends.

## Strengths

- The paper proposes a discretized codebook-based autoregressive approach for sleep foundation models, which is a promising direction given the strong findings of similar paradigms in related neural signal domains such as EEG and MEG. The authors effectively translate this formulation to sleep signals across different modalities and empirically demonstrate consistent improvements over several state-of-the-art sleep foundation models trained using alternative SSL- and contrastive learning-based objectives.

- The authors conduct extensive experimentation on the scaling behavior of the proposed architecture, demonstrating how performance varies with factors such as dataset size, model capacity, and training scale. These analyses provide useful insights into the scalability and robustness of the proposed approach.

- The paper is technically well written and maintains a clear overall flow. The authors provide sufficient implementation and experimental details to support reproducibility.

## Weaknesses

- The masking strategy proposed by the authors remains unclear. It lacks a proper mathematical formulation, making it difficult to interpret and relate to Figure 2, especially since sufficient intuition behind the approach is not developed. As the masking strategy constitutes one of the key novelty claims of the paper, this component requires significantly more clarity and formal explanation.

- The authors appear to use the SleepBench dataset configuration proposed in [1]. In that case, the detailed dataset description section becomes somewhat unnecessary, and the paper should explicitly clarify that the same experimental setup/configuration from [1] is being followed. Otherwise, readers may incorrectly interpret the dataset setup itself as an additional contribution of the paper.

- The evaluation metrics are reported inconsistently across experiments. For example, Table 1 reports only Cohen’s Kappa, whereas Table 2 switches to AUROC and AUPRC. Similarly, Table 5 reports Cohen’s Kappa for sleep staging tasks but AUROC and AUPRC for other downstream tasks. This lack of uniformity makes it difficult to consistently interpret and compare performance across tasks and experiments. Moreover, the selective reporting of different metrics across tables may give the impression that metrics are being chosen based on where the proposed approach performs more favorably. The authors should therefore either report all major metrics consistently across experiments, adopt a more uniform evaluation protocol, or provide stronger justification for why specific metrics are more appropriate for particular tasks.

- In Figure 3(b), the legend overlaps with the graph, affecting readability and overall presentation quality.

## Suggestions

- The authors should justify the proposed masking strategy in significantly greater detail, particularly by providing stronger intuition, clearer formulation, and discussion regarding its design choices and effectiveness.

- The authors should also better justify the overall evaluation strategy, including the choice of evaluation metrics for specific tasks and why those metrics are most appropriate in each setting.

- The paper currently lacks sufficient mathematical formulation, which is a critical component for a foundation model paper. Only two equations are provided in the appendix, leaving much of the architecture and training pipeline described only conceptually. The authors should develop a more rigorous mathematical understanding and formulation of the proposed architecture and objectives.

- Similarly, the overall methodology diagram is highly abstract and not sufficiently informative for a foundation model paper. Although the authors provide technical implementation details for reproducibility, it is equally important to include a clear and detailed methodology figure that helps readers intuitively understand the key components of the architecture and the overall data flow in an easily interpretable manner.

- The authors should also consider adding a more comprehensive ablation study to better understand the contribution of each individual component to the overall performance. Such analyses would improve the credibility of the proposed approach and are particularly important for foundation model papers to demonstrate the necessity and effectiveness of each design choice.

## References

1.) Shuai, Zitao, et al. "OSF: On Pre-training and Scaling of Sleep Foundation Models." arXiv preprint arXiv:2603.00190 (2026).


## Justification of Score

The paper presents a promising autoregressive foundation model for sleep signals and demonstrates strong empirical performance across multiple downstream tasks and benchmarks. The extensive scaling analysis and comparisons with prior sleep foundation models are valuable strengths of the work. However, the paper lacks sufficient clarity and mathematical formulation for key methodological components, particularly the masking strategy, which is central to the novelty claim. Additionally, inconsistencies in evaluation protocols and metric reporting reduce the overall clarity of the experimental analysis. Nevertheless, the paper explores a timely and important research direction and is likely to generate meaningful discussion at the workshop, especially given the growing interest in foundation models for biosignals.

---

### Official Review · Reviewer_Wzfy · 2026-05-19
**Good results, but the methodology and scaling analysis need clearer justification and evaluation.**

**Rating:** 5
**Confidence:** 5

**Review:**

This work presents Hypnos, a multimodal sleep foundation model trained using next-token prediction over residual vector-quantized tokens derived from eight physiological modalities. As demonstrated in the case studies, Hypnos achieves competitive performance compared to classical methods across different settings. However, several issues remain:

The tokenizer component requires more detailed explanation, particularly regarding how it handles missing modalities, which is claimed as a key contribution in Section I.
The training process of the Hypnos model needs further clarification. It is difficult to understand how data from different modalities interact or are integrated during training. The modality connections illustrated in the figure appear weak and insufficiently explained.
In the scaling analysis, the authors evaluate performance across different model parameter sizes, but they do not report or discuss the effect of varying input data sizes.